# River Basin Management Planning in the Republic of Ireland: Past, Present and the Future

**Sarpong Hammond Antwi [1],\*, Suzanne Linnane [1], David Getty [1] and Alec Rolston [2]**

1   Centre for Freshwater and Environmental Studies, Dundalk Institute of Technology,
    A91 K584 Dundalk, Ireland; suzanne.linnane@dkit.ie (S.L.); david.getty@dkit.ie (D.G.)
2   Goyder Institute for Water Research, Adelaide, SA 5005, Australia; alec.rolston@goyderinstitute.org
\*   Correspondence: Hammond.Sarpong@dkit.ie; Tel.: +353-892436273

**Abstract:** The River Basin Management Plan (RBMP) is an essential component of the European Union Water Framework Directive that details an integrated approach required to protect, improve and sustainably manage water resources. RBMP were intended to be produced for the periods 2009–2015, 2016–2021 and 2022–2027. However, after two years of delays in the development processes, the Republic of Ireland produced its first RBMP in 2010. The second RBMP cycle was also implemented in 2018 and is expected to run until the end of 2021 to give way to the third RBMP, whose consultation processes have been ongoing since December 2019. This paper contributes to the forthcoming RBMP by assessing stakeholders' perspectives on the second RBMP through a desk-based review and by conducting interviews with nine institutions (14 interviewees). The qualitatively analysed interviews reveal a broad spectrum of actors associated with water management and governance in the Republic of Ireland through a three-tier governance structure that has been delivered (with amendment) through the first two RBMPs. Organisations such as the An Fóram Uisce | The Water Forum, the Environmental Protection Agency, the Local Authority Waters, and the Agricultural Sustainability Support and Advisory Programme have responsibilities designated in the RBMPs to deliver improved water quality, integrated catchment management, community engagement and awareness-raising. Trust has also been building up among these organisations and other agencies in the water sector. Despite these responsibilities and progress made, the interviews identified communication lapses, ineffective collaboration and coordination among stakeholders and late implementation to be hampering the successful delivery of the second RBMP, in addition to significant pressures acting on water bodies from agricultural activities and urban wastewater treatment. Towards the third RBMP, the paper concludes that optimised water sector finance, enhanced and well-resourced communications, and improved stakeholder collaboration are needed to foster effective and efficient water services delivery and quality. More so, given the cross-cutting impact of the Sustainable Development Goals on water resources and the interconnected relations among the goals, the paper further recommends the integration of the SDGs in the various plans of actions and a co-benefits approach to derive the triple benefits from biodiversity, climate change initiatives and water quality measures.

**Keywords:** water framework directive; River Basin Management Plan; water resource management; water governance; stakeholders

## 1. Introduction

The European Union (EU), in response to the prevalent threats on water resources, developed and adopted the Water Framework Directive (WFD) in 2003 [1]. The WFD attempts to integrate a number of environmental policies and former directives (such as Nitrate Directives, Drinking Water Directives, Urban Waste Water Treatment Directive, etc.) and aims to pursue ecological goals for all water resources and investments in water protection measures and water ecology in the EU. It further provides direction toward

integrated water resource management and cross-cutting links to other EU legalisations that are relevant to the prevention, restoration and protection of fresh, coastal and transitional waters [1,2]. Currently, in excess of 110,000 water bodies across the EU are being managed under the WFD to reduce pollution and to improve water quality through functional water governance and management practices [3]. Despite its attempted holistic approach to improving water management, the WFD has been subjected to broad criticism. From a legal perspective, Santbergen [4] described the WFD as environmental legislation whose interpretation is very complicated because its ambivalent wording contradicts its very principles and objectives. The WFD also places a financial strain and intensely demanding timelines with some misunderstanding on key tenants of the technical and scientific aspects of the directive [5–7]. EU member states are further obliged to quantify the cost of the socio-economic and environmental effect of using water services under the WFD; however, article 9 (4) of the directive diffuses the same responsibility. The Article requires that "member states may not be in contravention of any principle if they choose in line with implementation not to comply with the cost recovery as far as it does not undermine the overall purpose and objective of the directive" [6,7]. Coordinating the WFD with policies such as the Common Agricultural Policy (CAP) and achieving compliance with Nitrate Directive are other problems identified to be thwarting efforts at reaching a good ecological status for water bodies [2]. These problems, according to Giakoumis and Voulvoulis [8], stem from the heavy influences from the European Parliament and environmental-based non-governmental organisations with different interests during the framework preparation. Nevertheless, a 2019 fitness check sanctioned under Article 19.2 of the WFD revealed that the framework was still flexible enough to deal with such threats, including emerging microplastics and pharmaceutical pollution and climate change, impacting water quality [3]. Thus, the WFD is still considered one of the most significant piece of legislation for water policy in Europe [8,9].

Under Article 14 of the WFD, EU member States are required to produce a River Basin Management Plan (RBMP) that sets out actions to regulate all member States' water quality concerns and also ensure the attainment of good ecological status for water bodies (rivers, lakes, estuaries and coastal waters) by 2027 through three RBMP cycles from 2009–2015, 2016–2021 and 2022–2027 [1,10].

## 1.1. RBMP in the Republic of Ireland

The WFD was written into law in the Republic of Ireland through the European Communities (Water Policy) Regulation 2003 (SI 722/2003) [5]. The Water Policy Regulations replaced a number of previous and existing legislative instruments aimed at improving water quality, including the European Communities Act 1972 Local Government (Water Pollution) Act 1977, the Quality of Bathing Water Regulations 1988, Local Government (Water Pollution) Act 1990, EPA Act (1992), Local Government Act 1994 and Waste Management Act 1996, the implementation of which had been fraught with management and governance challenges [11]. The WFD, since its legal adoption, has attempted to correct these by setting a benchmark for water management and governance by ensuring that water resources are grouped into catchments to enhance monitoring and attainment of good ecological status [11,12]. To achieve the required good ecological status means that all 4829 water bodies, comprising 111 coastal water bodies, 195 transitional waters, 818 lakes, 3192 rivers, 513 groundwater bodies and 15 artificial water bodies in the Republic of Ireland must reach a specific level that meets not only drinking and bathing needs but also agricultural, industrial and recreational needs as well as a healthy ecosystem that can support aquatic life [12,13]. In 2010, the Republic of Ireland produced its first RBMP, two years later than intended, to monitor, evaluate, and categorise surface and ground waters.

The delays affected the planning and implementation of the second plan, which consequently had a delivery period of four years instead of the required six-year duration. Although it is expected that the third RBMP will realign with the WFD timeframe of six years in 2021 [14], the delayed adoption and implementation of RBMPs has been common in other EU countries

for example Germany, Greece, Lithuania and Norway [15,16]. Dukelow [17], however, relates the delays in the Republic of Ireland to the Irish 2008 financial and economic meltdown and reforms in the water sector.

*1.2. First RBMP in the Republic of Ireland (2010–2015)*

The Environmental Protection Agency (EPA) first published its River Basin monitoring programme in 2006 and followed it up in 2007 with a report on significant water-management issues, after which a six-month public consultation was launched [18]. The outcome of the public consultation and Programmes of Measures (PoMS) was then published in December 2008, leading to the first RBMP formally adopted in 2009 [19]. The cost of the consultation and final production of the first RBMP was estimated at EUR 50 million [20]; even so, difficulties in differentiating the types of water resources, a single implementation approach and over-generalisation were some gaps that characterised the first RBMP [19]. Other major gaps identified in the first plan included poor development of assessment methods on the classification of ecological status, unclear methodology on cost recovery of water to domestic consumers and the absence of some quality elements (QEs) in the monitoring programme for lakes and coastal waters [21]. According to Earle and Blacklocke [22], the goals of the first plans themselves were unfeasible because the idea of RBMP was new in Europe. The plans' implementation also happened during the Irish economic crisis, which strained the needed fiscal resource for its implementation [14]. The absence of a single authority to oversee the plan with clearly defined responsibilities also restricted the opportunity for consultation and understanding between stakeholders, consulting authorities and various advisory councils, which was required to foster a culture of responsiveness [18,23]. In effect, the scientific basis of the plan became highly reliant on expert judgement [23]. A 2019 report by the EPA revealed that despite some improvement in water quality during the period that the first RBMP was in place, about 47.2% of water resources kept worsening. The report further disclosed that 44 out of 904 public water in 2016 could not meet the EU Drinking Water Regulation 2014 standards on pesticide and nitrate pollution [24]. (Figure 1).

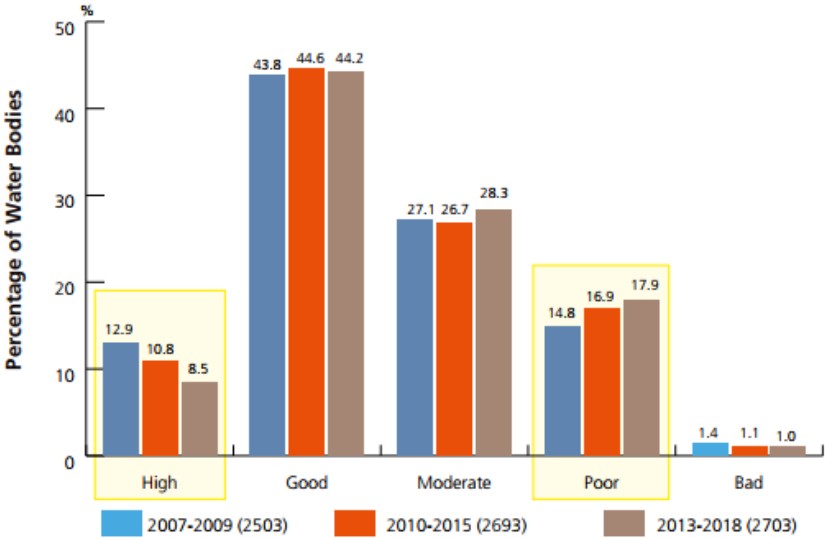

**Figure 1.** Percentage changes in water quality levels over a three-assessment period from 2007 to 2018 [24].

Overall, ecological assessment from 2013 to 2018 on 2703 surface water bodies and 514 groundwater showed 52.8% "satisfactory level" for surface water bodies while 47.2% remained "moderately poor" [24]. Although the first RBMP failed to reach its 13% national improvement in water quality status for the six-year period, it was a significant development in national water policy, leading to the establishment of eight River Basin

District (RBD) [10]. Licensing for urban-waste water discharges and agricultural regulations to protect water bodies were also introduced under the plan. The plan also enhanced water quality monitoring and implementation processes through established legal frameworks under European Communities Environmental Objectives (Groundwater) regulations 2010 (SI 9 of 2010) and European Communities Environmental Objectives (Surface Water) Regulations 2009 (SI 272 of 2009) [14].

*1.3. Second RBMP in the Republic of Ireland (2018–2021)*

The second RBMP was initiated in 2018, following a two-year delay, and is expected to run until the end of 2021. The second RBMP built upon lessons learned through the development, implementation and review of the first RBMP which identified gaps in public participation and governance processes required to meet objectives under the WFD. As a result, and to streamline national reporting requirements, the second RBMP combined separate river basins (i.e., Shannon, Western, South Western, Eastern and South-Eastern RBDs) into a single national river basin district. Two international RBDs—North Western and Neagh Bann—remained jointly managed with Northern Ireland (UK) to consolidate planning, monitoring and management [10]. A water quality indicator report for 2017–2019 under the period of the second RBMP revealed that 57% (1329) of river bodies attained a good biological quality while 43% (1002) remained in moderate quality [25]. While the report further indicated improvements in 2019, declines in high water bodies have not significantly halted through the first two RBMPs due to excess nutrients such as phosphorus and nitrogen, mainly from agriculture and wastewater [25]. This has further affected the number of pristine rivers across the country from 500 in the 1980s to 20 in 2020, indicating a 90% loss [9].

Aside the attempts to reverse the decrease in good water quality, the implementation of the second RBMP has introduced some reforms into the water sector. These include the formation of the Local Authority Waters Programme (LAWPRO) in 2018 to promote community engagement and raise awareness of water quality issues [26]. Other programmes introduced include the Agricultural Sustainability Support and Advisory Programme (AS-SAP) to promote sustainable agricultural practices in 190 targeted areas [27] and the 'Blue Dot Catchments Programme' to maintain and restore good water quality status [13]. An Fóram Uisce | The Water Forum was established as a statutory body under Water Services Act 2017 to also enhance democratic input into decision making in the water sector as part of RBMP implementation in the Republic of Ireland [28]. An investment of EUR 1.7 billion to deliver approximately 250 wastewater treatment projects and achieve 37% leakage reduction by 2021 are included within the second RBMP [13]. To further address governance issues, the three-tiered governance structure of the first RBMP was greatly enhanced (see Section 3) to provide clarity on the processes and actors involved in managing water river basins in the Republic of Ireland [10]. The governance structure aims at solving the cross-cutting challenges in the water sector coherently with a detailed consideration to agriculture, peat extraction and other identified water services issues [29].

Water governance and management under the second RBMP, nonetheless, have shown signs of susceptibility to external influences on water resources affairs due to the high tendency for a government through its state agencies and bodies to influence environmental affairs to suit its interest [30]. In addition, ineffective communication among relevant stakeholders, duplication of managerial roles and responsibilities and the ability of the RBMP to deal with the impact of drought on water resources and other recurring water resources challenges at catchment levels have all hampered the effectiveness of the second RBMP. Sustainable Water Network (SWAN), an umbrella NGO of Ireland's leading environmental organisations, concludes that the second RBMP lacks the ambitions needed to ensure water resources in Ireland stay clean and in good quality because of the reductions in water quality targets under the plan [30].

Although seminal contributions have been made on the implementation of WFD and the first RBMP in the Republic of Ireland [8,13,20], there has been limited scholarly

outputs on the second RBMP and its impact on governance and management of water resources except for two recent reports published by the Environmental Protection Agency in 2021 [31,32]. The reports were conducted using Experimental Governance Lens [32] and the Organisation for Economic Co-operation and Development (OECD) Water Governance Indicator Framework [31], with the findings from both reports emphasizing close policy and practical linkages between water, climate and biodiversity agendas and overall improvements in existing arrangements in the water sector. Considering the challenges with the implementation of RBMP and in view of limited assessment of the water governance and management implementation actions thus far, this paper assesses the second RBMP from a stakeholders' perspective. It does this by (i) identifying the successes and challenges with the second RBMP; (ii) cataloguing the expectations of stakeholders for the third RBMP, which can potentially improve the quality and effectiveness of policy measures required for the success of RBMP in the Republic of Ireland and then (iii) proposing suggestions that could positively contribute to achieving the objectives of the third RBMP for 2022–2027. Although the analysis presented in this paper focused on the Republic of Ireland, it is assumed that the findings are relevant to other European countries and regions where water sector planning, management and implementation challenges affects the overall achievement of good ecological status [15].

## 2. Materials and Methods

The findings in this study were derived from a mixed qualitative method using a desk-based review of the RBMP and key stakeholder interviews. The process involved:

- A review of water governance and management in the Republic of Ireland with a focus on the first and second RBMP to provide a baseline information and understanding of the governance processes and to validate emerging findings and evidence to inform policy and practice for the third RBMP. The review considered journal articles, annual reports and government policy papers in addition to submissions made by state and non-state institutions such as the Sustainable Water Network (SWAN); public consultation report on Significant Water Management Issues for Ireland published by the Department of Housing, Planning and Local Government (DHPLG) and various EPA reports in relation to water quality and the RBMP.
- Based on the approach of Gregory et al. [33], for selecting stakeholders for an interview, we identified and interviewed fourteen key stakeholders from nine institutions based on context and time, with multiple roles or positions related to the governance and management of water resources (Table 1). All the stakeholders interviewed had different degrees of expertise related to the management and governance of water resources in the Republic of Ireland, but due to difference in roles and responsibilities, some institutions had more than two stakeholders interviewed from within.

**Table 1.** Representative key stakeholders interviewed.

| Key Stakeholders Institutions | Total Number of Interviewees |
|---|---|
| Department of Housing and Local Government (DHLGH) | 1 |
| Environmental Protection Agency | 3 |
| Irish Farmers Association | 1 |
| Institute of Public Administration (IPA) | 1 |
| Local Authority Waters Programme (LAWPRO) | 3 |
| Maigue Rivers Trust | 1 |
| National Federation of Group Water Schemes (NFGWS) | 1 |
| Sustainable Water Network (SWAN) | 1 |
| Water Forum | 2 |

The open-ended qualitative interviews with the identified stakeholders aimed to gain insight into the implementation of RBMP (Supplementary Materials). Due to COVID-19 restrictions, the interviews were conducted remotely using Zoom. Nvivo 12 was used to

qualitatively analyse all interview responses which were coded into six themes (i.e., positive progress made under the second RBMP; significant challenges; Sustainable Development Goals; attaining the WFD objectives under the second RBMP; stakeholders' expectations; a general overview on water governance and management in the Republic of Ireland) and twenty-nine child nodes. The child nodes were derived from the themes to identify patterns and understanding from stakeholders' responses and to establish connections with their expectations for the third RBMP.

The number of stakeholders from state institutions and non-state institutions reflects the structure of the stakeholder community in the water sector. Moreover, the inclusion of Maigue River Trust, NFGWS and DHLGH shows a bottom (the impact of governance at catchment scale) to the top (national) analysis.

## 3. Results

Results from the desk-base review and stakeholder interviews were analysed to identify broad themes and understanding from the data gathered. These themes relate to the positive progress made under the second RBMP, significant challenges, Sustainable Development Goals, attaining the WFD objectives under the second RBMP, stakeholders' expectations and overview of water governance and management leading to recommendations for the third RBMP.

### 3.1. Positive Progress Made under the Second RBMP

The second RBMP, according to stakeholders, has contributed to efforts to improve local water quality and initiatives and imposed itself as the gateway to participatory governance and management of water resources under a three-tier governance structure (Figure 2). Although improved water quality takes time to manifest, the key institutions and actors involved in river basin management in the Republic of Ireland now work under a defined governance and management structure [10].

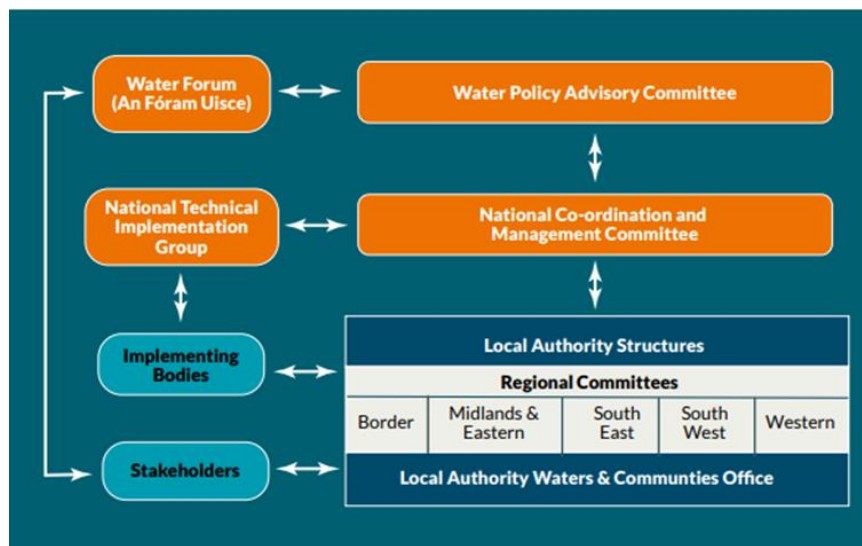

**Figure 2.** The three-tier governance structure for RBMP implementation [34].

The first tier of the governance structure exists under the auspices of The Water Policy Advisory Committee (WPAC), which is chaired by the Department of Housing, Planning, Community and Local Government. The WPAC monitors the implementation of the RBMP and offers policy advice and befitting recommendations to the Department of Housing, Planning, Community and Local Government, who provide the needed resources for implementing the plan [12,34]. With four meetings every year, WPAC also brings on board key policy-setting organisations together towards the preparation of RBMPs and to map up strategies and approaches towards achieving various objectives under the plan. An Fórum

Uisce | The Water Forum is a member of WPAC and represents its constituent stakeholders at this level.

The National Co-ordination and Management Committee (NCMC) under tier two also ensures that the measures outlined in the RBMP are strengthened through partnerships with various stakeholders and implementing bodies within the water sector. The EPA plays a significant role here as the responsible body that drafts the environmental objectives, manages catchment characterisation and produces the RMBP templates with input from local authorities [12]. In addition, the EPA has a history of being in tune with science-driven environmental management and does monitoring and reporting on the quality of environment, funding and coordination of environmentally related research under this tier [35].

Local authorities are also responsible for leading the implementation and enforcement of legislation on the ground and in encouraging public participation in decision making on RMBP under tier three [12]. The Local Waters and Communities Office, now the Local Authority Water Programme (LAWPRO), coordinates this with technical advice from the EPA. The establishment of LAWPRO and its emphasis on public participation and engagement stems from the failure of the first RBMP on decision-making processes and public participation. Commenting on the governance structure, former Minister for Housing, Planning and Local Government Eoghan Murphy TD stated that "It is to solve the cross-cutting challenges in the water sector coherently because of the detailed consideration it has given to other areas like agriculture, peat extraction and other identified water services issues" [29]. An analysis of the structure affirms this statement due to its consistent Catchment Management (ICM) features which further demonstrate attempts by the Republic of Ireland towards ICM adoption in managing catchments as implemented in Australia, South Africa, New Zealand and the USA, for instance [12,36]. ICM is regarded as a way of organising catchments as units for better understanding and management of ecosystem processes in a socio-economic and political context that offers communities an opportunity to turn input into sustainable natural resource management in their catchment. Local governments, communities and states have used the approach since 1988 for effective decision-making on catchments [37].

### 3.1.1. Institutional Set-Up

In contrast to the first RBMP, the second RBMP has resulted in structural changes with the implementation of new programmes and institutions. As agreed by all interviewees, a major success under the second plan is the introduction of active community engagement spearheaded by LAWPRO, which was not in the first plan. Within LAWPRO, the community and catchment scientist team engage communities to initiate actions to promote water quality. This has increased Tidy Town groups and community groups' focus on cleanliness and assisting with invasive species issues [26]. By the end of 2019, LAWPRO had reported a 62% completion of all its desk studies and held 111 community meetings and over 90 fieldwork assessments [38]. Another indicator of progress was the establishment of An Fórum Uisce | The Water Forum in 2018, which has since grown in stature, competence and capacity through continuous stakeholder engagement and contributions to the discussion on water policy at the national level [28]. There has also been significant catchment characterisation by the EPA catchment unit. The Agricultural Sustainability, Support and Advisory Programme (ASSAP) under the plan supports the implementation of best practice at the farm level in 190 Priority Areas for Action aimed at addressing agricultural pressures on water [27].

### 3.1.2. Participation and Collaboration

According to the interviewees, collaboration and participation among stakeholders in the water sector have also seen an improvement under the second plan, resulting in a gradual build-up of trust among agencies and working units in the water sector. The agricultural sector, which has been a significant source of pressure on water quality, has

seen improved level of interaction and discussion with other groups such as the Department of Agriculture, industries, individual farming bodies and the Dairy Sustainability Council. Thus, having local authorities liaise with farmers with input from the agricultural, processing and dairy industry, catchment scientists and ASSAP farm advisors have been a positive step. The Water Policy Advisory Committee, local authorities, Irish Water, EPA, Department of Agriculture and the Office of Public Works also meet four times a year to discuss emerging challenges, progress and measures to improve the water sector, which interviewees deemed as remarkable. Interviewees further identified LAWPRO's engagement with communities as a success, with an extra 120 to 130 new community groups that did not exist pre-2017 now involved in delivering action on the ground thanks to the Community Water Development Fund. As an interviewee said, *"The huge kind of network and stakeholder engagement status built up and the relationships that have been developed across all the public agencies and stakeholders have been absolutely unbelievable" (RI 4).*

### 3.1.3. Enhanced Governance and Management Processes

The first RBMP divided the Republic of Ireland into eight river basin districts, which resulted in disjointed and ineffective management [13]. The eight river basin districts have under the second RBMP been consolidated into one large river basin to monitor and implement actions effectively [10]. When asked about the pace of governance processes, the interviewees were unanimous in their view of improved coordination in the governance and management of water under the second plan due to structural changes under the second RBMP. The development of strategic tools and mechanisms to improve water quality and management through the Internet of Things (IoT) was also realised. For instance, the EPA Catchments Unit has a WFD app, enabling the local authorities to determine significant pressures. Catchment.ie webpage has also been established to disseminate science and stories about Ireland's water catchments and people's connections to their water. In addition, a host of other pollution potential and risk maps have been developed to identify critical sources. These web-based interfaces show where the most risks are likely to come from in terms of nitrogen and phosphorus on the landscape. One interviewee further revealed the on-going development of a risk assessment tool for hydro-morphology to enhance efficiency in the identification of physical characteristics and water content of water bodies across different catchments. These information dissemination portals and applications help foster water data management, providing decision-makers with feedback and indicators that are essential in planning and implementing policy decisions to improve water governance and management.

### 3.1.4. Awareness among Stakeholders and the Public

Improved awareness among stakeholders in the water sector and the general public was echoed by all interviewees. They alleged that the catchment works being carried out by LAWPRO, the stakeholder engagement by An Fórum Uisce | The Water Forum and publications by EPA and ASSAP in association with farmers had culminated in raising public awareness on water governance and management, which hitherto was limited. Evidence from case studies on local catchment groups in the Republic of Ireland from 2018 to 2020 revealed that catchment engagement and community events organized by LAWPRO enhanced the skill and capacity of river trust and catchments groups and improved their level awareness on catchment management [39]. Interviewees alleged that the various institutions across the water sector were now more aware of their roles and responsibilities, previously not well defined. Thus, the steady working relationship and defining roles and responsibilities contributed to awareness among stakeholders and the general public.

### *3.2. Significant Challenges under the Second RBMP*
### 3.2.1. Time and Financial Constraints

The implementation of the second RBMP, on the one hand, was hampered by a mixture of short timeframes and financial constraints. Stakeholder concerns in relation

to finance affirmed earlier findings by Boyle et al. [32], in which the Water Development Fund of EUR 225,000 opened to various community and voluntary groups involved in the protection and restoration of water at the catchment level in 2020 was deemed insufficient. The limited annual funding impacted catchment actions required to protect and improve local water quality and in delivering local benefits. Moreover, criticism of the plan as not being ambitious enough is traced to limited financial resources. The limited funding also transcends into urban wastewater treatment and constraints local authorities face in driving water quality improvements and protection functions, all of which have also been highlighted as a key challenge in the water sector [25]. A national funding strategy that spells out the funding of RBMP also remains unclear under the second RBMP. Some interviewees argued that financing of the RBMP is tied to a political will. At the same time, new units under the governance framework were set up with a limited connection to other units and tiers due to time constraints. Interviewees thus contended that all agencies and bodies had to learn how to work with each other under a limited period. COVID-19 also reportedly hindered the work done by community water officers in the summer of 2020, as most of their work is seasonal.

### 3.2.2. Governance Structure and Institutional Overlaps

The overall interviewees' response to the governance structure points to complications triggered by structural inefficiencies and overlaps in roles and responsibilities at both local and regional levels and among agencies despite the changes from the first RBMPP. Getting all bodies to contribute to the plan of action, according to interviewees, has been difficult because the plan and institutions within were not integrated enough. It was further alleged by an interviewee that the national coordination and management committee, for example, was dominated by engineers than environmental scientists who were either overburdened with responsibilities or lacked interest in water quality issues. The lack of clarity regarding the second RBMP's performance management, for instance, between various departments, the EPA and the agricultural sector, was also mentioned. Additionally, the proliferation of different agencies further hampered monitoring processes and the implementation of common actions and the identification of value for money. Taken together, these overlaps suggested that organizational structures and governance processes had not been efficiently coordinated.

### 3.2.3. Policy Coherence

A recurrent theme in the interviews was a sense among interviewees that policy coherence on what is important (i.e., water quality, flood relief, or agriculture) was not well distinguished. As a result, contradictions on who does what and to what extent remained a challenge under the second RBMP. Cited examples include LAWPRO's efforts at water quality management and improvement while, on the other hand, dairy expansion and agricultural activities continuously impact such efforts. While this may not be intentional, all interviewees agreed that it impedes efforts to reach the expected quality levels because the various institutions managing water appear to have no significant influence or direct power over those making decisions about agriculture. There are also a number of gaps in terms of implementation and supplementary measures needed, especially on urban and domestic wastewater, hydro-morphology, forestry and other pressures on water resources. The second RBMP was also identified as not being sufficiently integrated into other environmental laws and regulations. O'riordan et al. [31] posit that the absence of primary legislation to support the implementation the WFD also represents a challenge for the RBMP due to devolved responsibility on the enforcement of water abstraction, wastewater treatment directive and nitrates directives for instance which has also influenced EU infringement actions against the Republic of Ireland for non-compliance to WFD. The absence of primary legislation makes the court moderate on environmental breaches. In addition to the absence of a primary legislation, LAWPRO for instance has no enforcement

powers while a framework on accountability and code of conduct in the water governance arrangements remain unseen [31].

### 3.2.4. Communication

Although Section 3 highlights the positive progress made regarding public engagement and awareness-raising under the second RBMP, communication concerns were more widespread, particularly in identifying progress, areas of difficulties and in sharing learning among implementing bodies. Overall, interviewed stakeholders acknowledged that the General Data Protection Regulation (GDPR) on data access impacted the sharing of data openly even among various bodies in the governance structure. WPAC, for instance, meets frequently, yet records of their meetings are not detailed enough for other agencies to rely upon. Interviewees indicated that clear and early communication from the national coordinating committee, for instance, needed to feed into the action of local-level agencies, communities, and the general public, were not forthcoming. Thus, the lack of real-time data and the willingness to share information among institution and between implementation bodies served as a challenge under the plan. An interviewee stated that *"within the agricultural sector, the absence of preliminary figures regarding how much nitrogen needs to be removed from the agricultural system and targets on how much needs to be taken out by 2027 has not been communicated". ( R13)* These communication concerns further impacts the identification of data gaps, in monitoring and reviewing progress.

### 3.3. Sustainable Development Goals under the Second RBMP

Most of the measures and underlying objectives that constituted the design and implementation of the WFD were framed to address clean water and sanitation which fits into Goal 6 of the Sustainable Development Goals (SDG). The goal is about ensuring availability and sustainable management of water and sanitation for all [40]. Other related goals such as Goal 12: (Responsible Consumption and Production), Goal 13: (Climate Action) and Goal 14: (Life Below Water) also fit into the objectives and principles of the WFD. The Republic of Ireland and Kenya were the countries that facilitated the final phase of the intergovernmental negotiations for the acceptance of the goals; even so, the most striking results to emerge from the literature review and interview on the SDG showed limited public awareness of the goals in the Republic of Ireland [40,41]. Two recent assessments of the RBMP by the EPA did not consider the SDGs, which further affirms the limited recognition of them [31,32]. Although the SDGs were not explicitly stated in the second RBMP, some interviewees argued that they were linked to clean water objectives under the second plan. However, evident in the number of times the SDGs are referred to and the limited attention given in literature, it could be concluded that the RBMP did not try to achieve the SDGs the Republic of Ireland.

Moreover, considering all of the comments by interviewees, it appeared that the different units, bodies and departments deal with different issues and do not have a concerted approach to achieving the SDGs. As explained by an interviewee: *"RBMP ideally is to be a vehicle for the delivery of the SDG, but that is not clear. People have the perception of the goals as global issues and not local issues, but it is about local action for global action but that is not the perception in Ireland. There should be the linkage of what local communities are doing on water quality and how it is linked to environment locally and nationally, but that is not happening currently" (RI 05).* None of the interviewees could clearly identify with the success of the SDG under their organisation, although they recognize the need for the RBMP to help achieve the SDGs. The limited consideration and attention to the SDG, consequently, have implications on attaining not only the goals but, to a greater extent, the WFD by 2027 because the SDG and WFD objectives are parallel, and achieving either contributes to the other.

### 3.4. Attaining the WFD Objectives under the Second RBMP

Interviewed stakeholders expressed a high degree of uncertainty in achieving WFD by 2027. A lack of political will, underinvestment in the sector public participation and delays in implementing the RBMP were some factors attributed to the uncertainty by interviewees. Other factors included eutrophication (excess phosphorus and nitrogen in freshwater and estuarine), hydro-morphology (physical alterations and modifications of water bodies), agricultural activities, urban discharge and forestry activities. While these factors are prevalent not only in the Republic of Ireland but also many EU states and in England and Scotland, they require improved governance arrangements, approaches and active engagement with farmers who are pivotal in reducing these pressures and in the successful implementation of the WFD [42]. Commenting on Ireland's ability to attain the objectives of WFD, one of the interviewees said, *"At our current pace of progress? No way, it would be very difficult. I think it could be possible in some catchments. If there was a focus on the individual catchments, but given the way the management, governance is structured, and the continuation of the priority action areas, which are kind of piecemeal, I think it would be very difficult to achieve those targets across the board across the whole country" (RI 03).*

### 3.5. Stakeholders Expectations

The governance and management process under the second RBMP is considered experimental, with the expectation for some additions going into the third plan. Table 2 presents an overview of some key areas interviewees believe need improvement. Much of the expectation lies in implementing, monitoring and evaluating actions in agriculture and stakeholder engagements. Hydro-morphological pressures that affect over 329 rivers, 10 lakes, and six transitional water bodies require extra attention. In addition, poorly managed forest operations, peat extraction and activities which affect water quality would also require improved attention in the next plan coupled with investment in wastewater and leakage programs. There is also an expectation for the third plan to consider larger water bodies and not be limited to only 190 priority areas of action with greater emphasises on source protection. Primary legislation to support the implementation of the WFD in the Republic of Ireland is amiss, but to deal with significant pressures and activities that impact water bodies, the third plan is expected to produce clearly defined compliance approach to deal with polluters. A robust form of local and national environmental education that target farmers, the general public and schools in collaboration with state and non-state agencies is also expected under the third RBMP.

**Table 2.** Stakeholders' expectations for areas of improvement in the third RBMP.

| Areas of Improvement | Suggested Measures |
|---|---|
| Communication and coordination | Improve communication with the public, landowners, communities, farmers and implementing agencies. <br> Enhance communication between committees in the governance structure. For instance, sharing of minutes among committees could help avoid duplications and inefficiency in implementation. <br> Expansion of programmes in the agriculture sector such as ASSAP. ASSAP's working relationships with other agencies and bodies could help improve the focus from the productivity of farms and environmental biodiversity across the agricultural sector. |
| Governance structure | Greater collaboration between agencies and institutions is expected, particularly between the national coordinating management committee and the local authorities. <br> Distinguished guidelines on the roles of traditional local authorities and their environmental team and that of LAWPRO and ASSAP. <br> Distinction and coordination among local authorities and other institutions towards promoting implementation efficiency. <br> Policy coherence and robustness to improve and protect water resources. <br> Primary legislation to support the implementation of the WFD in the Republic of Ireland. |

**Table 2.** *Cont.*

| Areas of Improvement | Suggested Measures |
|---|---|
| Irish Water | Make Irish Water an integral part of RBMP implementation plans.<br>Improve action on wastewater, urban discharges and capital investment. |
| Monitoring and implementation | Greater emphasis on water protection activities by LAWPRO and ASSAP advisors and all 38 catchment scientists<br>Review of CAP and Nitrate Action Plan to ensure accountability and reward farmers upon delivering water quality, biodiversity and other climate benefits.<br>Deepen attention on pressures that affect water quality such as hydro morphology, forestry, invasive species and wastewater.<br>Synergies on actions required to promote forestry to derive the benefits of carbon capture.<br>Collaborative approach in implementing actions that has biodiversity, water and climate change benefits.<br>Mid-term progress monitoring and assessment of plan to track progress.<br>Current progress is regarded as slow, hence, focus on the 190 priority actions should be broaden and also focused on source protection. |
| Resource availability | Establish stream of funding to ensure that farmers can provide and ensure ecosystem services for the benefit of the environments.<br>Funding to ensure more priority areas are covered.<br>Catchment scientists should be available across all local authorities for efficient and effective monitoring and assessment across catchments.<br>The third plan should be released on time to avoid delays in implementation. |
| Stakeholder engagement | The third plan should see LAWPRO expand in areas like community engagement with wider stakeholders and broaden its scope on biodiversity and water in relation to wider communication and engagement.<br>Public participation and inputs from stakeholder should be key in the next plan.<br>Plan of action for all 46 attachment should be made available to the public and to all stakeholders. |

Although the community engagement by LAWPRO is generally perceived as good, stakeholders expect an expansion in community and stakeholder engagement. There are also expectations for mid-term progress monitoring and assessment of progress and collaborative approaches in implementing actions that have biodiversity, water and climate change benefits. As generally acknowledged by all interviewees; the three-tier governance structure is new and needs continuation; nevertheless, specific guidelines on the roles of traditional local authorities and their environmental team and that of LAWPRO and ASSAP, as well as greater collaboration between the national coordinating management committee and the local authorities, are among the expectations of stakeholders in the third RBMP.

## 4. Towards the Third RBMP

The RBMP challenges highlighted in Section 3.2 and expectations summarized in Table 2 are cumulative and reflect the impact of water governance and management under the second RBMP. Although this study identifies the current governance processes as being supported by a broad spectrum of stakeholders, the late implementation of the plan has affected the realization of its full impact especially on water quality, because it takes considerable time for quality standards and action to manifest. From the stakeholders' perspective and identified shortfalls, the study identifies and summarises the challenges with the second RBMP as being the following: finance to broaden priority areas and implementation of action plans, limited access to data and information on targets and progress, and inadequate coordination and collaboration between institutions and units as part of the governance process towards ensuring the planning and water quality protection as well as the SDG's becoming everyone's concern. Another challenge deduced from both desk review and interviews is related to innovation. Innovation through nature-based solutions, smart practices and state of the art technologies that could improve water management and service delivery and protect, improve, and sustainably manage the

environment were not sufficiently conceptualised in the RBMP processes. For instance, consensus towards smart metering for domestic water consumption, a national drought monitoring and early warning system, simplified administrative procedures through digitisation and extended public participation, as well as new tools and approaches to respond to sector needs, are either in their primary stages of development, implementation or not in existence. Another missing link is the multidisciplinary approach to addressing behavioural and societal values attached to water as a priority.

These shortfalls nevertheless offer a guideline in making adjustments in the upcoming third RBMP. The study complements stakeholders' expectations from interviews and review of literatures by offering the following suggestion: Firstly, to enhance effective and efficient communication, there could be the adaptation of digitisation and an online platform with a unified database that also allows internal and external communication to be fostered among all bodies at each level of the governance structure. Without sufficient access to data, information and communication among institutions and the public could impact decision-making and scientific-based approaches to improving water quality and reducing pressures. It could also impact shared learning and feedback of relevant information flow among stakeholders in the water sector.

Secondly, a co-benefits approach which is a strategy that conceptualizes both environmental benefits and social development in a single plan or policy framework could also be adopted to ensure that resources made available to implement the third RBMP yield the needed results at the catchment level. A co-benefits approach is essential given the interconnected nature of water to other sectors of the economy and the potential in triggering sustained socio-economic and infrastructural outcomes due to the strong relationship between co-benefits approaches and water resources [43,44]. Ürge-Vorsatz et al. [44] further argue that co-benefits could help resolve barriers faced by policymakers in implementing climate and environmental ambitions of which the water sector is an integral part. The Water Forum has already laid the foundation for co-benefits approach through a proposed Framework for Integrated Land and Landscape Management (FILLM), which, if implemented, could improve environmental outcomes in areas of water and ecosystem management towards meeting the country's environmental goals for climate adaptation and mitigation, biodiversity protection and water quality [45].

The governance structure is relatively new, from which ineffective coordination of the different agencies in the governance structure has resulted in fragmentation of actions and duplications of some roles and responsibilities [32]. Improved coordination, particularly around monitoring, implementation and engagement, are therefore needed for robust governance and management of water resources. A study on how to tackle diffuse pollution from agriculture in England and Scotland, for example, showed that institutional fragmentation hindered efforts among stakeholder in building trust and cooperation and in implementing stringent measures to tackle agricultural pollution in England. This was in contrast to Scotland where meaningful engagements of all stakeholders helped in tackling agricultural pollution [42]. Similarly, improved institutional coordination, monitoring and stakeholder engagements could also help solve agricultural pollution, which has a significant impact on water quality in the Republic of Ireland. When the institutions and units coordinate, collaborate and share resources including meeting minutes instead of working in silos, it could help complement the management and governance of water resources and avoid duplications and inefficiencies in implementation.

Additionally, to maximise the benefits of public participation and minimise the tendency of a "decide-announce-defend" posture, which mars the spirit of involvement, transparency and public participation, the next plan could outline strategic approaches towards public participation. Whereas there is no "one-size-fits-all" solution to public participation, lessons from France and Denmark and from other European countries on RBMP implementation through active participation are worth considering in the next plan because despite institutional legacies, active participation of decision- makers in the learning processes and knowledge production towards policy formulation and the clear

top-down and bottom-up approach to river basin institutions decision can influence high stakeholder participation and information flow [46]. To this effect, modern communication options, both virtual and physical, could be activated to simplify public engagement and participation processes under the third plan.

As stated earlier, the SDG's have not been significantly considered in environmental legislations and discourse over time in the Republic of Ireland. This has implications on developing coherent and relevant socio-ecological strategies and in building synergies towards tackling wastewater, water supply, sanitation and hygiene problems, which are tied to the goals [47]. It may also transcend into difficulties with managing the environment under the context of good health, responsive consumption and food production. Thus, given the cross-cutting impact of the SDG on water resources and the interconnected relations and trade-offs among the goals [40,41,48], we argue that it could be embedded into various actions of the third RBMP and also made explicit in various intended actions through coordinated implementation and improved awareness among stakeholders. This would improve the chances of attaining the SDG by 2030 and, to a more considerable extent, the WFD in 2027 through public awareness of the goals, multi-stakeholder partnerships and knowledge sharing towards the ultimate-water quality standards required under law.

Various comprehensive studies published by the EPA on water quality [25], bathing water quality [49] and the environment in general [9] have shown that water resources in the Republic of Ireland are not biologically healthy as they should be. Plans to improve and reduce pressures such as urban wastewater, diffuse pollution from agriculture and septic tank leakages which impact not only the biological quality of ground waters, rivers and lakes, and the quality of coastal water, bathing water and that of transitional (estuarine) are worth considering in the third plan. The plan could also consider key aspects of the environment, such as climate change and biodiversity and their interplay in water resource management and governance. This could be framed along with a gap analysis that espouses the progress, challenges and integrated approaches required to meet the 2027 water quality benchmark across all water resources in the Republic of Ireland. Lastly, the provision of sufficient funds is a requisite in delivering RBMP actions. In this regard, private funding options and external funding from the EU Green Deal and the European Agricultural Fund for Rural Development, for instance, could be explored to help meet the fiscal requirement for infrastructural revamp, deployment of state-of-the-art techniques and equipment in water supply, including research, and the cost of fixing leakages. Moreover, in broadening priority areas, emphasis should be placed on rural development, rewetting of peatlands and deployment of more catchment scientists under LAWPRO to serve community needs and help in the building of resilience in the water sector.

## 5. Conclusions

This paper contributes to the governance and management of water resources by highlighting stakeholders' perspectives of the second River Basin Management Plan (RBMP) in the Republic of Ireland. It identifies the water governance and management processes under RBMP as being supported by a broad spectrum of stakeholders through a three-tier governance structure that clarifies the processes and actors involved in the water sector. Institutions such as An Fóram Uisce | The Water Forum, Agricultural Sustainability Support and Advisory Programme (ASSAP) and the Local Authority Waters Programme (LAWPRO) have been effective under the plan. However, stakeholders argue that it is unrealistic to assert that the Republic of Ireland could meet the 2027 water quality benchmark based on progress under the RBMP. The reasons for this include the late implementation of the plan, communication lapses and ineffective collaboration and coordination among stakeholders. Agriculture and forestry activities, peat extraction, eutrophication and hydro-morphology were also significant pressures on water resources. Stakeholders' expectations for the upcoming RBMP suggest the need for a centralised information system to implement effective and efficient communication among stakeholders. There is also a need for increased financial investment to broaden priority areas and the integration of the Sustainable

Development Goals in catchments actions towards water quality improvement. The paper further recommends the need for co-benefits approaches to derive the triple benefit from biodiversity, climate change initiative and water quality measures in the third RBMP. Although the context of this paper is limited to the Republic of Ireland, its findings could be replicated to suit the local context in other European countries and regions who aim at implementing integrated river basin management policies.

**Supplementary Materials:** The following are available online at https://doi.org/10.5281/zenodo.4946714.

**Author Contributions:** Conceptualization, S.H.A., A.R., S.L. and D.G.; writing—original draft preparation, S.H.A., writing—review and editing, S.H.A., S.L., D.G. and A.R.; supervision, S.L., D.G. and A.R.; funding acquisition, S.L., D.G. and A.R. All authors have read and agreed to the published version of the manuscript.

**Funding:** The Dundalk Institute of Technology funded this research through Landscape funding.

**Informed Consent Statement:** Informed consent was obtained from all subjects involved in the study.

**Acknowledgments:** Special thanks to all interviewees who accepted to be part of the study. Special thanks to Suzanne Smith of DkIT for her assistance with the Nvivo software.

**Conflicts of Interest:** The funders had no role in the design of the study; in the collection, analyses or interpretation of data; in the writing of the manuscript or in the decision to publish the results. The authors declare no conflict of interest.

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
