# Peer review of "River Basin Management Planning in the Republic of Ireland: Past, Present and the Future"

_water, doi:10.3390/w13152074_

Round 1

Reviewer 1 Report

Dear Authors,

your manuscript is interesting and presents the situation of the Republic of Ireland in a very comprehensive way.

As you can see in the attached file, I have only minor hints.

One of the major drawbacks of such kind of paper is, as also pointed out by you in the Conclusions, the very local focus. In revising the text, I suggest improving the literature review, to provide some examples of countries having a similar situation to the Republic of Ireland, therefore improving the overall impact of your work.

Author Response

Dear Reviewer, Thanks for your valuable feedback which has greatly improved the manuscript. All concerns have been addressed for fit for scientific purposes. 

Point 1: I'm struggling to see how this section constitutes 'results' as currently phrased and worded.  I assume this is the desk study of literature/policy papers.  However, it reads as 'fact' or givens and presents a largely uncritical perspective on the material reviewed.  Far too many claims are just accepted and presented as reality.  Needs to be far more nuanced and exploring differences within the materials reviewed.  It certainly is unclear that this is the result of the desk study.

Response 1:

Thanks for your comments, but the claims here are indeed facts attributed to the second RBMP. The three-tier structure, as indicated by interviewees, is relative. Critics and recommendations are further highlighted under subsequent themes under institutional set-ups and participation and collaboration. We admit there was too much information that was not significantly necessary. We have since summarized the section

Point 2: and who says / on what basis that this is better than 8?

Response 2:

The eight river basin districts were fraught with institutional challenges at a point where RBMP was relatively new. As referenced in the sentence, grouping all river basin districts into one helps improve easy monitoring and effective implementation of the plan of action.

Point 3: This is a useful table, but a list of possibilities does not automatically equate to an integrated system of governance.  Ideally, you would provide some critical commentary on this listing - is it in the next section?

Response 3:

Thanks for your comment. The heading has been reframed as “Suggested Measures” to reflect your suggestion. The table is a snapshot of all concerns raised by stakeholders in the interview processes. The following section discusses the most critical among the rest that significantly impacts water quality and is achievable within the 3rd RBMP life cycle.

Point 4: i. or mentioned by your interviewees? if not why are you so confident that these will be improvements?

  1. does this arise from your interviews / desk study?

Response 4: The section is a summary of all shortfalls from the interview and desk study.

Reviewer 2 Report

This paper offers some insights into the experiences of, and future possibilities for, WFD governance processes and implementation in the Republic of Ireland.  It is based on a desk study and a focussed set of interviews with key stakeholders.  

The paper works well for the most part in providing an account of the existing problems and possible improvements.  The link to SDGs offers a new dimension to the WFD debates.  But there are several areas that need attention.   The methods are barely described so it is difficult to assess how the findings were determined.  The presentation of the desk study findings lacks a critical voice and it is not always clear whose voice is being heard when discussing the interviewees. The analysis and recommendations are too often normative and hopeful, and lack grounding in the evidence.   A stronger narrative needs to be developed which offers a more critical view of the material to the reader to reveal more nuanced insights arising from the evidence to reveal the complexities and contradictions of policy and practice. 

The annotated file indicates these main areas and also additional minor elements that need attention.

Author Response

All comments have been addressed. Attached is the revised manuscript 

Round 2

Reviewer 2 Report

Most of the changes I identified have been made, but there are one or two areas where not changes early on in the paper have been made nor any justification given to reject the reviewer suggestion.  

I was unable to open a document from the journal website which presumably explained your response to the reviewer why no changes were made, so I have reiterated a few of my previous comments.   

I still find the discussion section to be unclear what has come from the lit and what has been derived from the interviews.  The preceding table of suggested improvements is also not discussed clearly.  Some minor edits here would help integrate the table, the lit and the interviewee findings.  

I look forward to the final version.

Author Response

All concerns have been addressed with the full responses attached. 
